# Risk factors for conversion to thoracotomy in patients with lung cancer undergoing video-assisted thoracoscopic surgery: A meta-analysis

**Siyu Wang, Hong Yan◉\*, Jun Wen, Zitong Zhou, Jialan Xu**

School of Nursing, Chengdu University of Traditional Chinese Medicine, Chengdu, Sichuan Province, People's Republic of China

\* yhcq2@163.com

**Data Availability Statement:** All relevant data are within the manuscript and its Supporting Information files.

## Abstract

### Objective

To systematically evaluate the risk factors of conversion to thoracotomy in thoracoscopic surgery (VATS) for lung cancer, and to provide a theoretical basis for the development of personalized surgical plans.

### Methods

CNKI, Wanfang, VIP, CBM, PubMed, Cochrane Library, Web of Science, and Embase databases were searched by computer from the establishment of the database to March 2024. Relevant studies on the risk factors of conversion to thoracotomy in VATS for lung cancer were searched. Two reviewers independently performed literature screening, data extraction, and quality evaluation, and Stata16.0 software was used for data analysis.

### Results

A total of 14 studies were included in this study, with a total sample size of 10605, and a total of 11 risk factors were obtained. Mate analysis showed that, Age $\geq$ 65 years old [$OR(95\% CI) = 2.61(1.67,4.09)$], male [$OR(95\%CI) = 1.46(1.19,1.79)$], BMI(Body Mass Index) $\geq$ 25 [$OR(95\%CI) = 1.79(1.17,2.74)$], tuberculosis history [$OR(95\%CI) = 7.67(4.25,13.83)$], enlarged mediastinal lymph nodes [$OR(95\%CI) = 2.33(1.50,3.06)$], lung door swollen lymph nodes [$OR(95\%CI) = 6.33(2.07,19.32)$], pleural adhesion [$OR(95\%CI) = 2.50(1.93,3.25)$], tumor located in the lung Upper lobe [$OR(95\%CI) = 4.01(2.87,5.60)$], sleeve lobectomy [$OR(95\%CI) = 3.40(1.43,8.08)$], diameter of tumor $\geq$ 3.5cm [$OR(95\%CI) = 2.13(1.15,3.95)$] associated with lung cancer VATS transit thoracotomy.

### Conclusions

Age $\geq$ 65 years old, male, BMI $\geq$ 25, tuberculosis history, enlarged mediastinal lymph nodes, lung door swollen lymph nodes, pleural adhesion, tumor located in the lung Upper lobe, sleeve lobectomy, diameter of tumor $\geq$ 3.5cm are risk factors for conversion to

**Funding:** The funding source of our paper is the Chengdu University of Traditional Chinese Medicine 2023 University-level First-class Course Construction Project-Community Nursing and Construction of digital curriculum resources and system for community nursing based on knowledge mapping [2022ZLGC26].

**Competing interests:** The authors have declared that no competing interests exist.

thoracotomy during VATS for lung cancer. Clinicians should pay attention to the above factors before VATS to avoid forced conversion due to the above factors during VATS. Due to the number and limitations of the included studies, the above conclusions need to be validated by additional high-quality studies.

## Trail registration

The protocol was registered into the PROSPERO database under the number CRD42023478648.

## Introduction

Lung cancer is one of the most prevalent malignant neoplastic diseases and the leading cause of cancer mortality. It is estimated that approximately two million new cases of lung cancer and 1.76 million lung cancer-related deaths occur globally each year [1]. Additionally, lung cancer has a high incidence and mortality rate in China. According to the International Agency for Research on Cancer Global Cancer Statistics, the number of new cases and deaths of lung cancer in China is about 820000 and 710000 in 2020 [2]. Surgery is the best treatment for lung cancer, and common surgical modalities include traditional thoracotomy and video-assisted thoracic surgery (VATS) [3]. Compared with traditional thoracotomy, VATS has the advantages of smaller surgical wounds, less postoperative pain, and fewer postoperative complications, and has gradually become the preferred surgical treatment for lung cancer [4]. However, lung cancer VATS requires not only the removal of cancerous lung tissue but also the dissection of surrounding lymph nodes, which is still at risk in lung cancer patients during the process of VATS due to the small surgical incision and narrow surgical field [5]. Studies have reported that the conversion rate of VATS to thoracotomy in lung cancer is 2%~23% [6]. Conversion to thoracotomy during VATS will inevitably result in an enlargement of the surgical wound, an extension of the surgical procedure and hospital stay, and an increased likelihood of a poor prognosis. At present, there are more and more studies on the risk factors of conversion to thoracotomy during VATS for lung cancer, but the results are not completely consistent. Therefore, this study aims to systematically evaluate the risk factors of conversion to thoracotomy during VATS for lung cancer by searching various databases, to help clinicians find high-risk patients as early as possible and develop personalized coping plans in advance.

## Materials and methods

### Literature search strategy

The related articles on the influencing factors of conversion to thoracotomy during VATS for lung cancer were searched by the combination of subject words and free words in CNKI, Wanfang, VIP, CBM, PubMed, Cochrane Library, Web of Science, and Embase databases. The search time limit was from the establishment of the database to March 2024. The search terms were lung cancer, pulmonary cancer, pulmonary neoplasm, video-assisted thoracic surgery, VATS, thoracoscopic Surgery, conversion to thoracotomy, and thoracotomy, retrieved articles were simultaneously screened for their references. The complete search strategy is shown in S1 File. Furthermore, databases such as ClinicalTrials.gov and the World Health Organization's ICTRP were also searched for ongoing clinical trials.

## Literature inclusion and exclusion criteria

Inclusion Criteria: (1) The type of study was a case-control study or cohort study; (2) The subjects were lung cancer patients undergoing VATS; (3) Any study that includes risk factors for conversion to thoracotomy during VATS for lung cancer; (4) The outcome was conversion to thoracotomy during VATS for lung cancer.

Exclusion Criteria: (1) The language of the literature was not Chinese or English; (2) Case studies, reviews, conference reports, etc. (3) Repeated searches and publications; (4) Studies that did not report relevant data; (5) Inaccessible or retracted literature; (6) Patient has a combination of other thoracic cancers, such as esophageal cancer.

## Literature screening and data extraction

Two reviewers independently performed literature screening and data extraction. Firstly, the literature retrieved in different databases was exported to EndNoteX9 for literature duplication removal. The title and abstract of the literature were preliminarily screened, and the full text was downloaded and read. In case of disagreement, two reviewers could negotiate or ask a third reviewer to repeat the above process and make a judgment. The final data extracted mainly included the first author, publication year, country, surgical method, sample size, rate of conversion to thoracotomy, and risk factors.

## Literature quality assessment

The content of the literature was subjected to a detailed evaluation by two independent reviewers to ascertain the potential for bias. The quality of case-control studies and cohort studies was evaluated by the Newcastle-Ottawa Scale (NOS) [7]. The total score of the scale was 9 points, 0–3 points for low-quality literature, 4–6 points for medium-quality literature, and 7–9 points for high-quality literature. When the evaluation results were inconsistent, two reviewers could negotiate or a third literature evaluator could be asked to evaluate.

## Statistical methods

Stata16.0 was used for meta-analysis, and odds ratio (OR) and 95% confidence interval (CI) were used to combine the effect sizes. The heterogeneity test was analyzed by Q test and $I^2$ quantification. If $I^2 \leq 50\%$ and $P \geq 0.1$, the heterogeneity of each included literature was small, and the fixed effect model was used to combine the effect size. If $I^2 > 50\%$ and $P < 0.1$, the heterogeneity of each included literature was large, and the random effect model was used to combine the effect size. The test level of the combined effect size was set as $\alpha = 0.05$. Sensitivity analysis was performed using Stata software. A funnel plot was used to detect publication bias when the number of included studies was more than 10.

## Result

### Results of the literature search

A total of 2547 relevant literature were retrieved from the databases, including 1477 Chinese literature and 1070 English literature. No ongoing studies or other unpublished literature were identified during the course of this investigation. A total of 14 articles were included after literature duplication removal, primary screening, and secondary screening. The literature screening flow chart is shown in Fig 1.

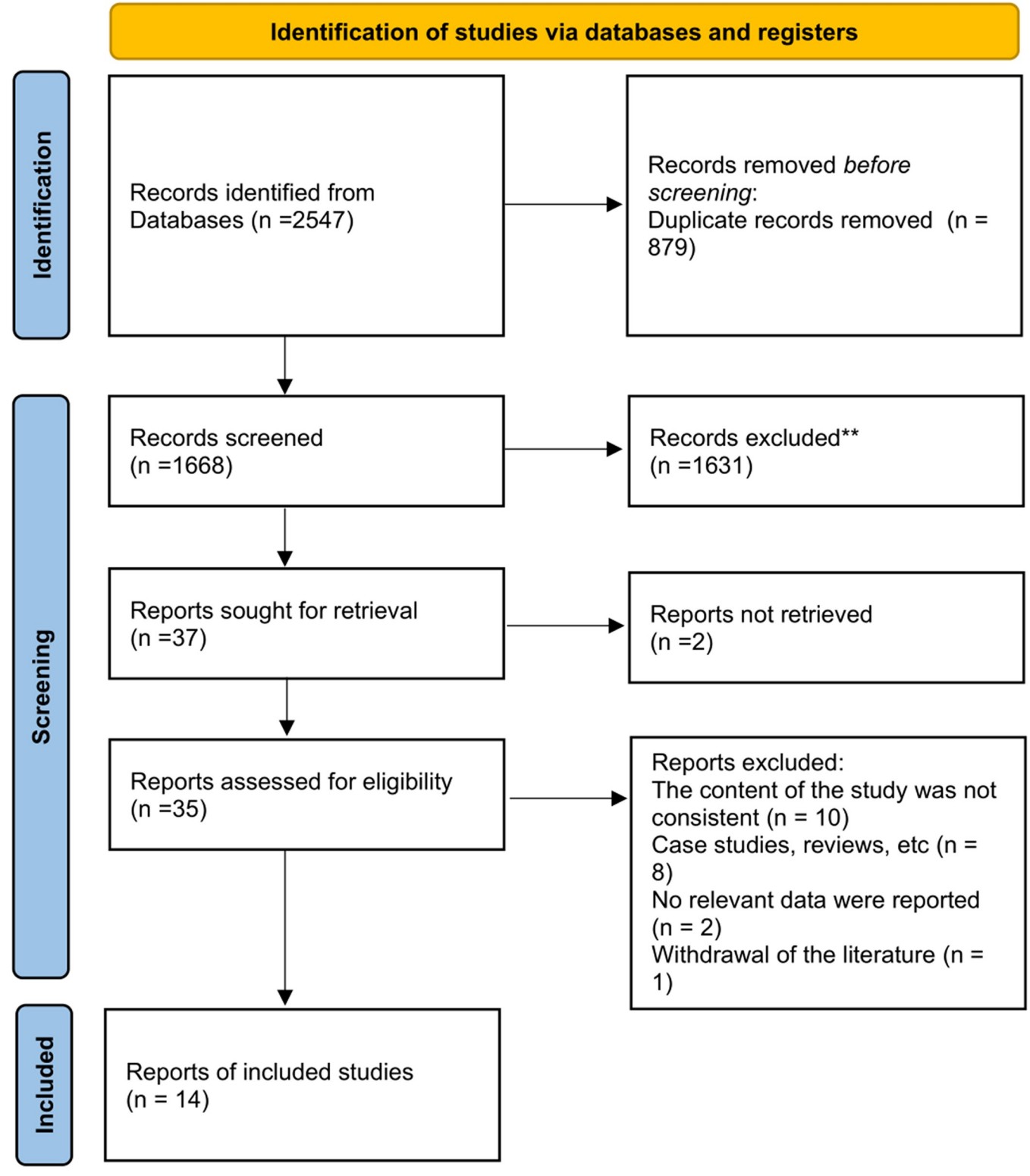

**Fig 1. Flow chart of literature screening.**

**Table 1. Basic information table of included literature.**

| Included literature | Country | Surgical methods | Sample size | | Rate of conversion to thoracotomy | Risk factors | Quality score |
|---|---|---|---|---|---|---|---|
| | | | Thoracoscopy group | Transferred thoracotomy group | | | |
| Jing Liu 2022 [8] | China | VATS radical lobectomy | 420 | 43 | 9.3% | a, d, e, f, h, g | 7 |
| Xin Liu 2023 [9] | China | VATS lobectomy | 96 | 24 | 20.0% | a, g, j | 6 |
| Fu-Qian Wu 2020 [10] | China | VATS radical lobectomy | 146 | 80 | 35.4% | a, d, g, h | 8 |
| Jie-Sheng Zhou 2023 [11] | China | VATS radical resection of lung cancer | 470 | 120 | 20.3% | d, g, h, i | 8 |
| Wei-Hua Li 2017 [12] | China | VATS lobectomy | 378 | 45 | 10.6% | e, f, i | 7 |
| Hai Li 2018 [13] | China | VATS radical lobectomy | 117 | 65 | 35.7% | a, b, g, h, j | 7 |
| An- Ge Chen 2021 [14] | China | VATS radical resection of lung cancer | 97 | 29 | 23.0% | a, d, g, h | 6 |
| Lim 2017 [15] | South Korea | VATS lobectomy | 180 | 55 | 23.4% | a | 6 |
| Kim 2017 [16] | South Korea | VATS lobectomy and segmental resection | 51 | 38 | 42.7% | a, h | 6 |
| Li 2017 [17] | China | VATS Single lobectomy | 459 | 16 | 3.4% | a, c, d, g | 6 |
| Gabryel 2021 [18] | Poland | VATS anatomical segmentectomy | 897 | 105 | 10.5% | e, g | 7 |
| Chen 2021 [19] | China | VATS anatomical segmentectomy | 997 | 180 | 15.3% | c, g, j | 6 |
| Bongiolatti 2019 [20] | Italy | VATS lobectomy | 4197 | 432 | 9.3% | a, b, k | 7 |
| Fourdrain 2022 [21] | France | VATS anatomical segmentectomy | 7100 | 843 | 10.6% | c, i, k | 8 |

Note: "a" is age, "b" is gender, "c" is BMI, "d" is history of pulmonary tuberculosis, "e" is mediastinal lymph node enlargement, "f" is hilar lymph node enlargement, "g" is pleural adhesion, "h" is tumor location, "I" is sleeve lobectomy, "j" is tumor diameter, "k" is clinical stage

## Basic information of the included literature

All the included studies were case-control studies, including 7 Chinese articles and 7 English articles, involving a total of 11 risk factors. The lowest quality score of the literature was 6 points and the highest was 8 points. Table 1 shows the basic information of the included literature. The particular NOS score attributed to each study is presented in S1 Data.

## Results of meta-analysis

The results of the meta-analysis for each risk factor are presented in Table 2, while forest plots for each risk factor are provided in S1 and S2 Figs. The particular data extracted for each study are presented in S2 Data.

## Patient's risk factors

The patient's risk factors included age, gender, BMI, and history of tuberculosis. Eight studies [8–10, 13–17] reported on the association between age and conversion to thoracotomy in VATS for lung cancer, of which five studies [8, 9, 13, 15, 17] had consistent criteria and could be combined. The combined results of fixed-effects models indicated that the risk of conversion to thoracotomy in patients aged ≥65 years was 2.61 times higher than that in patients aged <65 years. The combined results of fixed-effects models from 2 studies [13, 20] showed that the risk of conversion to thoracotomy during VATS in male patients was 1.46 times higher than that in females. The association between BMI and mid-stage open chest in VATS for lung cancer was reported in three studies [17, 19, 21] of which two studies [17, 19] had consistent

**Table 2. Results of the meta-analysis.**

| Risk Factors | Number of Included studies | Results of heterogeneity test | | Model of effect | OR (95%CI) | P value for the total effect |
|---|---|---|---|---|---|---|
| | | P | I² | | | |
| Age ≥ 65 | 5 | 0.116 | 46% | Fixed | 2.61 [1.67, 4.09] | <0.001 |
| Male | 2 | 0.962 | 0% | Fixed | 1.46 [1.19, 1.79] | <0.001 |
| BMI≥25 | 2 | 0.556 | 0% | Fixed | 1.79 [1.17, 2.74] | 0.008 |
| History of tuberculosis | 4 | 0.301 | 18% | Fixed | 7.67 [4.25, 13.83] | <0.001 |
| Mediastinal lymph node enlargement | 3 | 0.312 | 14% | Fixed | 2.33 [1.50, 3.06] | <0.001 |
| Swollen hilar lymph nodes | 2 | 0.153 | 51% | Random | 6.33 [2.07, 19.32] | 0.001 |
| Pleural adhesions | 9 | 0.235 | 24% | Fixed | 2.80 [2.27, 3.46] | <0.001 |
| The tumor is located in the upper lobe of the lung | 7 | 0.305 | 16% | Fixed | 4.01 [2.87, 5.60] | <0.001 |
| Sleeve lobectomy | 2 | 0.106 | 62% | Random | 3.40 [1.43, 8.08] | <0.001 |
| Tumor diameter ≥3.5cm | 2 | 0.998 | 0% | Fixed | 2.13 [1.15, 3.95] | 0.016 |
| TNM clinical stage I | 2 | 0.002 | 89% | Random | 1.42 [0.68, 2.97] | 0.352 |

Note: "cm" is centimeter; "TNM clinical stage I" is lung cancer cells located in the primary site without metastasis

delineation criteria and could be combined. The combined fixed-effects model demonstrated that patients with a BMI≥25 exhibited a 1.79-fold increased risk of conversion to thoracotomy during VATS compared to those with a BMI < 25. Five studies [8, 10, 11, 14, 17] reported on the association between a history of tuberculosis and conversion to thoracotomy during VATS for lung cancer. Following sensitivity analyses, four of these studies [8, 10, 14, 17] could be combined for a meta-analysis. The combined results of the fixed-effects models demonstrated that patients with a history of tuberculosis exhibited a 7.67-fold increased risk of conversion to thoracotomy during VATS in comparison with those without a history of tuberculosis.

**Risk factors for peri-tumor tissues.** Risk factors for peri-tumor tissues include enlarged mediastinal lymph nodes, enlarged hilar lymph nodes, and pleural adhesions. The findings from the fixed-effects models of three studies [8, 12, 18] demonstrated that individuals exhibiting mediastinal lymph node enlargement exhibited a 2.33-fold elevated risk of conversion to thoracotomy during VATS for lung cancer compared to those without mediastinal lymph node enlargement. The combined random-effects model results from two studies [8, 12] demonstrate that the risk of thoracotomy during VATS for lung cancer is 6.33 times greater for patients with hilar lymph node enlargement compared to those without. The combined fixed-effects model results of nine studies [8–11, 13, 14, 17–19] demonstrated that patients with pleural adhesions exhibited a 2.80-fold increased risk of conversion to thoracotomy during VATS for lung cancer compared to those without pleural adhesions. After correction by the cut-and-patch method, the result was 2.5 times higher.

**Risk factors for tumor characteristics and mode of resection.** Risk factors for Tumor characteristics and resection modalities included Tumor location, sleeve lobectomy, Tumor diameter, and tumor clinical stage. The aggregated results of the fixed-effects model of the seven studies [8–11, 13, 14, 16] demonstrated that patients with Tumors situated in the upper lobes of the lung exhibited a 4.10-fold elevated risk of conversion to thoracotomy during VATS for lung cancer in comparison to those with Tumors in the non-upper lobes of the lung. The combined random-effects model results of the two studies [11, 12] demonstrated that patients undergoing sleeve lobectomy exhibited a 4.10-fold increased risk of conversion to thoracotomy during VATS compared to lobectomy. Three studies [9, 13, 19] reported on the association between Tumor diameter and intermediate chest opening in VATS for lung cancer. Of

these, two studies [9, 13] had consistent delineation criteria and could be combined. The combined results of the fixed-effects model demonstrated that patients with Tumor diameters≥3.5 cm exhibited a 2.13-fold increased risk of conversion to thoracotomy during VATS compared to those with diameters < 3.5 cm. The combined results of the random-effects models of the two studies [20, 21] demonstrated no statistically significant difference in the effect of Tumor clinical stage on the conversion to thoracotomy during VATS for lung cancer.

**Sensitivity analysis and publication bias.** A sensitivity analysis was conducted for each risk factor using the one-by-one exclusion method and the interconversion of fixed-effects and random-effects models. The results of the one-by-one exclusion method indicated that there were some discrepancies between the study conducted by Zhou [11] and other studies examining the history of tuberculosis. After the exclusion, the $I^2$ value decreased from 75% to 18%. The sensitivity analysis of the history of tuberculosis is shown in Fig 2. The findings of the model transformation method demonstrated that the results of clinical staging stage I exhibited significant variability across different models, indicating a potential for unreliable outcomes. The sensitivity analysis results of the other factors were relatively stable. The Egger test indicated the possibility of publication bias in the pleural adhesion factor (P≤0.05), and the effect sizes were adjusted through the use of the cut-and-patch method. The adjusted OR was found to be 2.50 (95% CI: 1.93–3.25), demonstrating no significant change compared to the pre-adjustment period. This finding suggests that the publication bias did not exert a notable impact on the meta-analysis results. Consequently, the results obtained were deemed to be

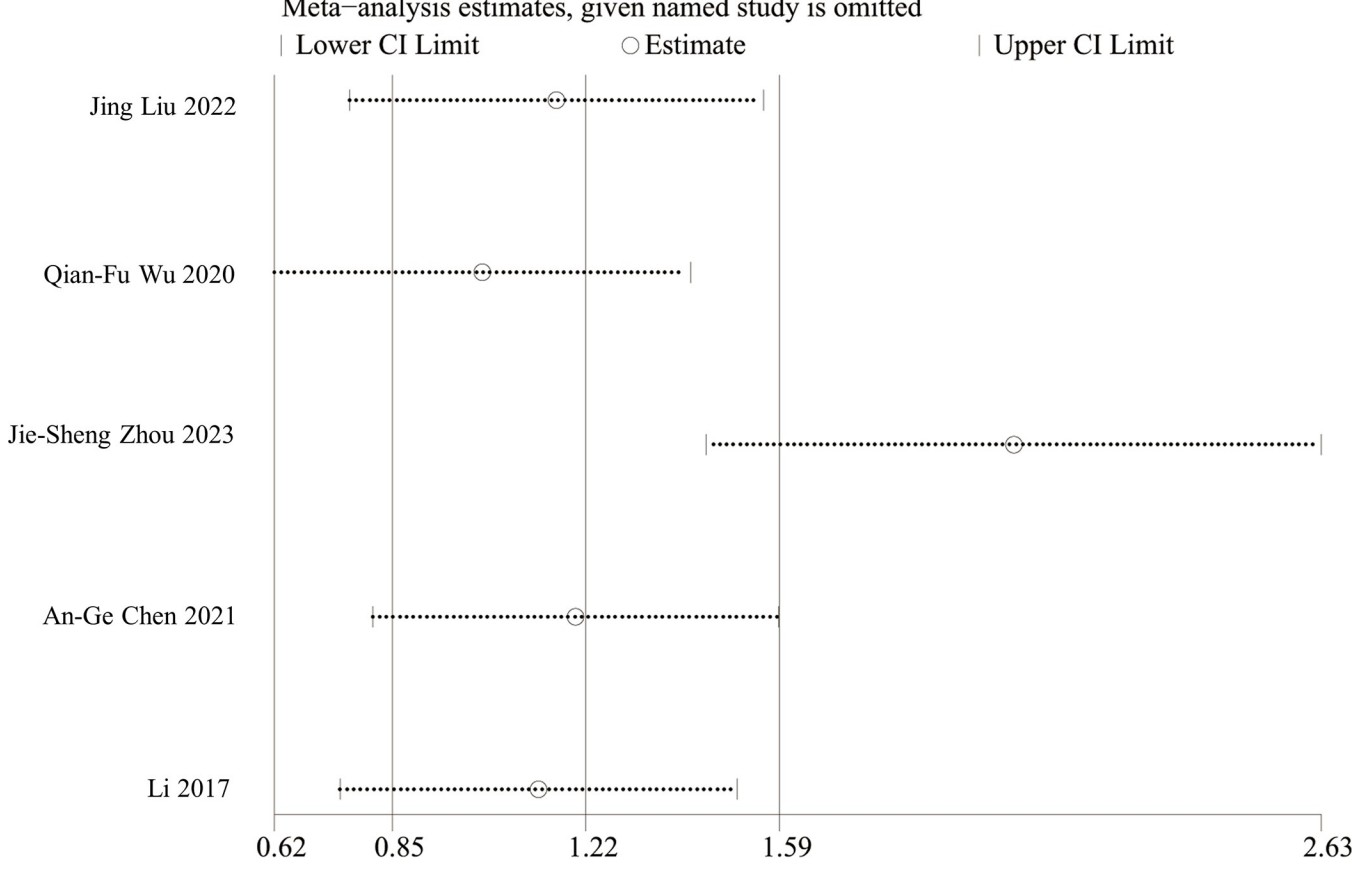

**Fig 2. Sensitivity analysis of pulmonary tuberculosis history.**

**Table 3. Sensitivity analyses and publication bias results.**

| Risk Factors | fixed-effects model | | random-effects model | | Egger test |
|---|---|---|---|---|---|
| | OR (95%CI) | P值 | OR (95%CI) | P值 | |
| Age ≥ 65 | 2.61 [1.67, 4.09] | <0.001 | 2.73 [1.47, 5.09] | 0.002 | 0.418 |
| Male | 1.46 [1.19, 1.79] | <0.001 | 1.46 [1.19, 1.79] | <0.001 | — |
| BMI≥25 | 1.79 [1.17, 2.74] | 0.008 | 1.79 [1.17, 2.74] | 0.008 | — |
| History of tuberculosis | 7.67 [4.25, 13.83] | <0.001 | 7.72 [3.99, 14.93] | <0.001 | 0.648 |
| Mediastinal lymph node enlargement | 2.33 [1.50, 3.06] | <0.001 | 2.30 [1.43, 3.71] | 0.001 | 0.653 |
| Swollen hilar lymph nodes | 6.04 [2.78, 13.12] | <0.001 | 6.33 [2.07, 19.32] | 0.001 | — |
| Pleural adhesions | 2.80 [2.27, 3.46] | <0.001 | 2.91 [2.26 3.74] | <0.001 | 0.019 |
| The tumor is located in the upper lobe of the lung | 4.01 [2.87, 5.60] | <0.001 | 4.10 [2.81, 5.96] | <0.001 | 0.443 |
| Sleeve lobectomy | 3.06 [1.86, 5.04] | 0.001 | 3.40 [1.43, 8.08] | <0.001 | — |
| Tumor diameter ≥3.5cm | 2.13 [1.15, 3.95] | 0.016 | 2.13 [1.15, 3.95] | 0.016 | — |
| TNM clinical stage I | 1.60 [1.27, 2.00] | <0.001 | 1.42 [0.68, 2.97] | 0.352 | — |

Note: "—" is result not shown

stable. Additionally, the remaining risk factors did not exhibit any substantial publication bias (P>0.05). The sensitivity analyses and publication bias for each factor are shown in Table 3.

## Discussion

With the ongoing advancement of thoracoscopic surgery, the incidence of conversion to thoracotomy during VATS for lung cancer has declined since 2016 [22]. However, some lung cancer patients still require conversion to thoracotomy intraoperatively, which significantly increases the complexity of the procedure and the surgical trauma, and has a profound impact on the patient's postoperative recovery [23]. It is therefore essential to select the appropriate surgical technique by the risk factors associated with VATS.

### Impact of patient's factors on conversion to thoracotomy during VATS for lung cancer

Some studies have found that thoracoscopic lung surgery requires longer operation time than open chest surgery [24], and patients aged 65 years or above exhibit a significantly shorter operative tolerance period due to the effects of advanced age, which increases the risk of undergoing intermediate thoracotomy due to their inability to tolerate the procedure for an extended period. VATS surgery exceeding 360 minutes may elevate the likelihood of postoperative complications. Therefore, VATS in older patients should be expedited to completion within 360 minutes if feasible.

Men are the main group of smokers [25]. Long-term smoking exposes lung tissue to the chemical substances produced by tobacco repeatedly, which causes inflammatory reactions and cell damage, leading to a more complicated surgical process [26], and the risk of conversion to thoracotomy is greater. A study demonstrated that non-smoking resulted in a reduction in intraoperative blood loss; however, the length of time spent abstaining from smoking did not affect intraoperative blood loss [27]. Furthermore, a reduction in intraoperative blood loss lowers the probability of conversion to a thoracotomy procedure during VATS. Consequently, it is recommended that surgeons request patients to abstain from smoking during the perioperative period, even in the absence of cessation after the initial diagnosis of the Tumor. This will have a beneficial impact on the surgical procedure and the postoperative recovery period [28].

During thoracoscopic surgery, overweight or obese patients with BMI≥25 are more likely to develop atelectasis and hypoxemia due to the accumulation of subcutaneous fat, which leads to limited diaphragmatic movement and thoracic expansion, and decreased lung compliance [29]. To reduce adverse symptoms, they are forced to convert to thoracotomy. When dealing with overweight or obese patients with a BMI ≥25, the surgeon should develop a personalized surgical plan based on the patient's weight.

Studies have found [30] that pulmonary tuberculosis can cause severe calcification and adhesion of lymph nodes, which seriously interferes with the visual field under thoracoscopic surgery. To expand the surgical field, conversion to thoracotomy is necessary. Accordingly, when encountering patients with a history of tuberculosis, it is imperative to conduct a meticulous assessment of lymph node calcification and adhesions before selecting an optimal surgical procedure.

## The influence of surrounding tissues on conversion to thoracotomy during VATS for lung cancer

Lymph nodes are generally distributed along the direction of blood vessels. When various factors lead to the enlargement of mediastinal and hilar lymph nodes, the clarity of local anatomical structures will be affected, causing serious interference with the field of view under thoracoscopic surgery and increasing the risk of vascular injuries, which will lead to the conversion to thoracotomy procedure during VATS for lung cancer [31]. According to the study of Matsuoka et al. [32], the conversion rate of thoracoscopic surgery to thoracotomy due to pleural adhesion was about 8%-20%. It is difficult to treat patients with pleural adhesion under thoracoscopy, which can easily lead to massive bleeding when separating the adhesion tissue, and the risk of conversion to thoracotomy is higher [33]. The most common cause of conversion to thoracotomy procedure during VATS is vascular injury, followed by anatomic causes such as adhesions and the presence of large or sticky lymph nodes. Before surgery, the surgeon should perform a careful assessment of the difficulty of VATS surgery in a patient based on imaging, to determine whether open chest surgery should be performed.

## The impact of tumor characteristics and resection methods on conversion to thoracotomy in VATS for lung cancer

Tumor diameter is an important indicator for choosing surgical methods, but the exact value is still unclear. It is generally believed that VTAS can be chosen when the Tumor diameter is small, and thoracotomy is chosen when it is large [34]. In this study, a Tumor diameter of 3.5 cm is considered to be the threshold for thoracotomy. For lung cancer patients with Tumors greater than or equal to 3.5 cm, the decision to perform open thoracotomy should be made with careful consideration.

It has been determined that the arterial vasculature in the upper lobes of the lungs is characterized by increased thickness, branching complexity, and the presence of intricate surrounding tissue, which collectively renders the surgical excision of this region susceptible to the potential for vascular damage [35]. The surgical procedure, termed sleeve lobectomy, has been designed to facilitate the conservation of the greatest possible amount of native lung tissue; nevertheless, the inherent technical complexity of the operation may potentially result in inadvertent damage to adjacent tissues and blood vessels [36]. Such factors increase the probability of conversion to thoracotomy during the operative period.

The impact of clinical staging on conversion to thoracotomy during VATS for lung cancer has been inconsistent across different studies. While Bongiolatti's [20] study demonstrated no statistically significant association between staging and conversion, Fourdrain's [21] study

yielded contrasting results. The meta-analysis of the aggregated data revealed that the discrepancy in the impact of the clinical stage on the likelihood of conversion to open chest surgery in VATS for lung cancer was not statistically significant. Given that the Fourdrain study had an elevated NOS score, we believe that further investigation is warranted in this area. We expect that further high-quality literature will corroborate this view.

## Limitations and prospects

First of all, the sample size of each included literature is quite different, and the surgical methods of each study are also different, which may cause some heterogeneity. Secondly, in this study, different included literature may have different classification criteria for a factor, and some unpublished literature may not be included, resulting in a small number of included literature for some factors, and the publication bias detection cannot be completed, which may have a certain impact on the accuracy of the results. Finally, the types of studies included in this study were case-control studies, and more prospective, multi-center, and high-quality literature should be included for further verification and to obtain more risk factors of conversion to thoracotomy after VATS for lung cancer.

## Supporting information

**S1 File. Literature search strategy.**
(DOCX)

**S1 Data. Newcastle-Ottawa Scale score.**
(XLSX)

**S2 Data. Detailed data in the literature.**
(XLSX)

**S3 Data. Literature search results.**
(XLSX)

**S1 Fig. Forest plots of risk factors.**
(TIF)

**S2 Fig. Forest plots of risk factors.**
(TIF)

## Author Contributions

**Conceptualization:** Siyu Wang, Hong Yan.

**Investigation:** Siyu Wang, Jun Wen.

**Methodology:** Siyu Wang.

**Resources:** Jun Wen.

**Software:** Siyu Wang, Zitong Zhou.

**Validation:** Zitong Zhou.

**Visualization:** Jialan Xu.

**Writing – original draft:** Siyu Wang.

**Writing – review & editing:** Hong Yan.

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
