## [Decision Letter · Decision Letter 0]

16 Aug 2024

PONE-D-24-25985Risk factors for conversion to thoracotomy in patients with lung cancer undergoing video-assisted thoracoscopic surgery: A meta-analysisPLOS ONE

Dear Dr. Yan,

Thank you for submitting your manuscript to PLOS ONE. After careful consideration, we feel that it has merit but does not fully meet PLOS ONE’s publication criteria as it currently stands. Therefore, we invite you to submit a revised version of the manuscript that addresses the points raised during the review process.

We look forward to receiving your revised manuscript.

Kind regards,

Luca Bertolaccini, M.D., Ph.D.

Academic Editor

PLOS ONE

Journal Requirements:

3. We note that you have referenced (unpublished) on page 14, which has currently not yet been accepted for publication. Please remove this from your References and amend this to state in the body of your manuscript: (ie “Bewick et al. [Unpublished]”) as detailed online in our guide for authors

4. Please respond by return e-mail with an updated version of your manuscript to amend either the abstract on the online submission form or the abstract in the manuscript so that they are identical. We can make any changes on your behalf.

Additional Editor Comments:

The reviewers have emphasised issues that require a careful and thorough revision of the manuscript.

No commitment to publication can be made at this point.

Reviewers' comments:

Reviewer's Responses to Questions

**Comments to the Author**

1. Is the manuscript technically sound, and do the data support the conclusions?

Reviewer #1: No

Reviewer #2: Yes

2. Has the statistical analysis been performed appropriately and rigorously? 

Reviewer #1: No

Reviewer #2: Yes

3. Have the authors made all data underlying the findings in their manuscript fully available?

Reviewer #1: Yes

Reviewer #2: Yes

4. Is the manuscript presented in an intelligible fashion and written in standard English?

Reviewer #1: No

Reviewer #2: No

5. Review Comments to the Author

Reviewer #1: The authors conducted a meta-analysis regarding the risk factors for conversion from VATS to thoracotomy. Despite it should an argument of interest, I think there are some important criticisms in the manuscript:

-the concept of this analysis is questionable. The goal of a metanalysis should be to clarify the role of procedure or risk factor that present controversial results in literature or that regards a small number of patients in separate studies. In this paper, the authors considered 11 risk factors separately, also including only 2 studies in some risk factors. I think that this methodology is more suitable for a narrative review that for a meta-analysis.

-Inclusion and exclusion criteria are not clear. It is not clear if in the separate analysis were included only significant studies on the risk factor and the others did not consider it.

-English should be revised.

-Discussion is very poor and its sounds as a repetition of the results, with small contribute on clinical impact of the study.

Reviewer #2: Thank you for submitting your manuscript titled “Risk factors for conversion to thoracotomy in patients with lung cancer undergoing video-assisted thoracoscopic surgery: A meta-analysis" for consideration. I appreciate the opportunity to review this informative work.

Your meta-analysis addresses an important topic in thoracic surgery, synthesizing data on risk factors for conversion from VATS to open thoracotomy in lung cancer patients. Strengths of the manuscript include a comprehensive literature search, appropriate statistical methods, and clinically relevant findings that could inform surgical planning and patient counseling.

To further improve the quality and impact of your work, I have the following comments and suggestions:

1. The conclusion in the abstract seems quite definitive. Would it be more appropriate to qualify some statements, given the limitations you have noted?

2. In the Literature Search Strategy subsection, please provide the complete search strategy for at least one database, including all search terms, Boolean operators, truncation symbols, and any limits applied. You could also consider including the full strategies for all databases as an appendix. This would improve reproducibility.

3. Were any steps taken to identify unpublished or ongoing studies? This may include searching clinical trial registries, conference proceedings, or contacting experts in the field. Additionally, were any attempts made to contact study authors for missing or unclear data? Please include these details to strengthen your methodology and adhere to PRISMA guidelines.

4. The heterogeneity for some risk factors (e.g., hilar lymphadenopathy) is quite high. Have you considered performing subgroup analyses or meta-regression to explore potential sources of this heterogeneity?

5. The forest plots for each risk factor would be useful additions. Can these be included, perhaps as supplementary material if space is limited in the main text?

6. Could you provide more detail on how the NOS scores influenced your interpretation of the results? For instance, did you consider giving more weight to higher quality studies in your analysis?

7. Did you observe any temporal trends in conversion rates or risk factors when comparing earlier versus more recent studies, given the evolution of VATS techniques?

8. Could you expand the discussion on clinical implications? For example, how might surgeons use this information to reduce conversion rates? Are there any preoperative interventions that could mitigate some of these risk factors?

9. Consider discussing the potential impact of the observed publication bias on your results for factors where this assessment was possible.

10. The manuscript would benefit from a careful review for language and grammar. For example, there are some awkward phrasings in the Results section that could be clarified.

Thank you again for the opportunity to review your work, and I look forward to seeing a revised version of the manuscript.

6. PLOS authors have the option to publish the peer review history of their article (what does this mean?). If published, this will include your full peer review and any attached files.

Reviewer #1: No

Reviewer #2: **Yes: **Savvas Lampridis

---

## [Author Response · Author response to Decision Letter 0]

25 Sep 2024

Responds to the reviewer’s comments:

Reviewer #1: 

1. In this paper, the authors considered 11 risk factors separately, also including only 2 studies on some risk factors. I think that this methodology is more suitable for a narrative review that for a meta-analysis.

Response: We are grateful for this comment. We contend that quantitative metrics offer a more objective perspective, despite the limited scope of the included literature. The risk factors for which the literature was limited were also subjected to comprehensive analysis and discussion.

2. Inclusion and exclusion criteria are not clear. It is not clear if the separate analysis were included only significant studies on the risk factor and the others did not consider it.

Response: We are very grateful for this comment. The meaning in the original article may not have been clearly expressed. All studies that included risk factors for conversion to thoracotomy during VATS will be included. The wording has been amended accordingly.

3. English should be revised.

Response: We are very grateful for this comment. The English of this article has been revised from beginning to end and hopefully will be compliant with PLOS ONE standards.

4. Discussion is very poor and it sounds as a repetition of the results, with a small contribute to clinical impact of the study.

Response: This commentary offers valuable insight that enhances the quality of this discussion. The portion of the discussion that replicated the results section was removed. Furthermore, the discussion was expanded to include the various options that physicians should consider when faced with each risk factor. For instance, patients aged 65 years and older should be operated on within 360 minutes. In cases of tumors larger than 3.5 cm, the surgeon is responsible for determining whether a thoracotomy is necessary before the operation. Similarly, patients with a history of tuberculosis must undergo imaging in advance, and the most suitable surgical procedure should be chosen based on their lymph node calcification and adhesion. Further detailed revisions are presented in the manuscript.

Special thanks to you for your good comments. 

Reviewer #2: 

1. The conclusion in the abstract seems quite definitive. Would it be more appropriate to qualify some statements, given the limitations you have noted?

Response: We have corrected according to the Reviewer’s comments. The abstract has included the statement ‘Due to the number and limitations of the included studies, the above conclusions need to be validated by additional high-quality studies’.

2. In the Literature Search Strategy subsection, please provide the complete search strategy for at least one database, including all search terms, Boolean operators, truncation symbols, and any limits applied. 

Response: We are grateful for your proposal. The full search strategy for PubMed has been uploaded as Supplementary file S1.

3. Were any steps taken to identify unpublished or ongoing studies? This may include searching clinical trial registries, conference proceedings, or contacting experts in the field. Additionally, were any attempts made to contact study authors for missing or unclear data? Please include these details to strengthen your methodology and adhere to PRISMA guidelines.

Response: We are grateful for this question. A comprehensive search of ClinicalTrials.gov and the WHO ICTRP database was conducted, and no evidence of ongoing studies was identified. This information has been incorporated into the methodology section. Furthermore, as no data were absent from the included literature, we did not contact the authors for additional information.

4. The heterogeneity for some risk factors (e.g., hilar lymphadenopathy) is quite high. Have you considered performing subgroup analyses or meta-regression to explore potential sources of this heterogeneity?

Response: We are grateful for your counsel. The possibility of subgroup analysis has been considered in advance, but the limited availability of high-heterogeneity risk factors in only two of the included literature sources precludes their analysis. We have discussed the potential factors contributing to the high degree of heterogeneity observed, which may include the size and characteristics of the study sample, the geographical location of the research setting, and other factors.

5. The forest plots for each risk factor would be useful additions. Can these be included, perhaps as supplementary material if space is limited in the main text?

Response: We are grateful for your invaluable guidance. In consideration of the article's length, all forest plots have been provided as supplementary documentation(S3, S4).

6. Could you provide more detail on how the NOS scores influenced your interpretation of the results? For instance, did you consider giving more weight to higher-quality studies in your analysis?

We would like to express our gratitude to the reviewers for their valuable comments. The findings of the high-quality literature were largely concordant with our own, except the domain of clinical staging, where a single piece of literature offered an opposing perspective. In light of the higher NOS score awarded to the study, we believe that this is still an area worthy of further investigation. We hope that further high-quality literature will corroborate our viewpoint. This has now been incorporated into the discussion section.

7. Did you observe any temporal trends in conversion rates or risk factors when comparing earlier versus more recent studies, given the evolution of VATS techniques?

Response: This advice is beneficial for our investigation. The studies included in the article did not demonstrate any temporal trends in conversion rates or risk factors. However, a review of the literature revealed that the incidence of conversion to thoracotomy surgery during VATS decreased after 2016 in comparison to the period preceding this. This has been addressed in the discussion section.

8. Could you expand the discussion on clinical implications? For example, how might surgeons use this information to reduce conversion rates? Are there any preoperative interventions that could mitigate some of these risk factors?

Response: This commentary offers valuable insight that enhances the quality of this discussion. The portion of the discussion that replicated the results section was removed. Furthermore, the discussion was expanded to include the various options that physicians should consider when faced with each risk factor. For instance, patients aged 65 years and older should be operated on within 360 minutes. In cases of tumors larger than 3.5 cm, the surgeon is responsible for determining whether a thoracotomy is necessary before the operation. Similarly, patients with a history of tuberculosis must undergo imaging in advance, and the most suitable surgical procedure should be chosen based on their lymph node calcification and adhesion. Further detailed revisions are presented in the manuscript.

9. Consider discussing the potential impact of the observed publication bias on your results for factors where this assessment was possible.

Response: We are very grateful for this comment. The effect sizes were adjusted using the trim and fill method, and the adjusted OR was 2.50 (95% CI: 1.93-3.25), which showed no significant change compared with the pre-adjustment period. This indicates that publication bias had little effect on the results of the meta-analysis and that the results obtained were stable.

10. The manuscript would benefit from a careful review of language and grammar. For example, there are some awkward phrasings in the Results section that could be clarified.

We are very sorry for the incorrect English expression. We have revised the language section from start to finish.

Special thanks to you for your good comments.

---

## [Decision Letter · Decision Letter 1]

22 Oct 2024

Risk factors for conversion to thoracotomy in patients with lung cancer undergoing video-assisted thoracoscopic surgery: A meta-analysis

PONE-D-24-25985R1

Dear Dr. Yan,

We’re pleased to inform you that your manuscript has been judged scientifically suitable for publication and will be formally accepted for publication once it meets all outstanding technical requirements.

Kind regards,

Luca Bertolaccini, M.D., Ph.D.

Academic Editor

PLOS ONE

Additional Editor Comments (optional):

Reviewers' comments:

Reviewer's Responses to Questions

**Comments to the Author**

1. If the authors have adequately addressed your comments raised in a previous round of review and you feel that this manuscript is now acceptable for publication, you may indicate that here to bypass the “Comments to the Author” section, enter your conflict of interest statement in the “Confidential to Editor” section, and submit your "Accept" recommendation.

Reviewer #2: All comments have been addressed

2. Is the manuscript technically sound, and do the data support the conclusions?

Reviewer #2: Yes

3. Has the statistical analysis been performed appropriately and rigorously? 

Reviewer #2: Yes

4. Have the authors made all data underlying the findings in their manuscript fully available?

Reviewer #2: Yes

5. Is the manuscript presented in an intelligible fashion and written in standard English?

Reviewer #2: Yes

6. Review Comments to the Author

Reviewer #2: Thank you for submitting your revised manuscript. I have carefully reviewed it along with your responses to the previous round of comments. I commend you on the thorough approach you have taken in addressing the concerns raised.

The revisions you have made have significantly improved the quality and clarity of your manuscript. Your clarification of the methodology, including details about the search strategy for ongoing studies, strengthens the scientific rigor of your work. The expanded discussion section now provides more comprehensive clinical implications. Your inclusion of limitations, particularly regarding the inability to conduct subgroup analyses due to limited data, demonstrates a balanced and critical approach to your research.

Overall, I believe the manuscript now presents a comprehensive, well-analyzed study with clear clinical relevance. Your work should make a useful contribution to understanding risk factors for conversion to thoracotomy in video-assisted thoracoscopic surgery for lung cancer.

Thank you again for your submission, and I wish you all the best in your research endeavours.

7. PLOS authors have the option to publish the peer review history of their article (what does this mean?). If published, this will include your full peer review and any attached files.

Reviewer #2: **Yes: **Savvas Lampridis

---

## [Editor Report · Acceptance letter]

6 Nov 2024

PONE-D-24-25985R1 

PLOS ONE

Dear Dr. Yan, 

I'm pleased to inform you that your manuscript has been deemed suitable for publication in PLOS ONE. Congratulations! Your manuscript is now being handed over to our production team.

Kind regards, 

on behalf of

Dr. Luca Bertolaccini 

Academic Editor

PLOS ONE